# The Image of Monks and the Monastic Community in the Latest Russian Cinematography

Joanna Kozieł 🆔

Institute of Linguistics, Faculty of Humanities, The John Paul II Catholic University of Lublin,
20-950 Lublin, Poland; joanna.koziel@kul.pl

**Abstract:** This article is an attempt to analyse how monastic communities are presented in the latest Russian cinematography. It is an issue that has not been well researched so far, because scholars tend to focus primarily on broadly understood religious films. Considering the achievements of the last 25 years, two selected films were analysed in terms of the aforementioned themes, both at the level of visual and verbal representations, as well as at the level of interpretation. In addition, the attitude of the Russian Orthodox Church to individual visions of the artists was taken into account. The research results indicate that one can identify the most popular motifs in films about monastic life. Moreover, the monastery itself most often becomes a place of refuge and at the same time a place of transformation for the heroes. In recent years, the Russian Orthodox Church has had a significant impact on artists' visions.

**Keywords:** monasticism; contemporary Russian cinema; Russian orthodox church; religious series; religious film

## 1. Introduction

Despite constant changes, spirituality and religion are topics that still play an important role in the modern world. This thesis is also confirmed by culture, where content is constantly being rediscovered and presented in various ways. The subject of this article is one of the aspects of the issue, namely the use of motifs related to monastic life in cinematography. Through the analysis of individual titles, an attempt was made to indicate what elements or problems the artists pay attention to and the image of the monk that is created in popular culture. At the same time, the study takes into account the attitude of the Russian Orthodox Church (ROC) to cinematography and to specific visions of film artists. The subject of religious cinema has been of interest to researchers for a long time, and the theologians Amédée Ayfre and Henri Agel are considered the precursors (Agel and Ayfre 1961).

It is a truism to say that religious themes have appeared in cinema from the very beginning of its existence. This also applies to Russian cinema, where elements of Orthodox worship have been appearing since the end of the 19th century. This is confirmed, for example, by Alfred Fedetsky's film, which captured the transfer of the icon of Our Lady from Kurjazh to Kharkiv. Neither the negative attitude of the Holy Synod during the tsarist era nor the years of forced atheism during the Soviet era changed the artists' interest in the subject. The presence of spiritual culture has not been wiped off the screen, and in this context, it is impossible not to mention the masterpieces of Andrei Tarkovsky. Currently, Russian artists also use themes related to religion, presenting them in different aspects, using different genre conventions or means of expression. It can be noted that this is part of

a trend common to world cinema in recent years (Junke 2018, p. 11)—issues of spirituality and religion are returning to the screens of Russian viewers more and more often[1]. It is worth adding that in contrast to the beginning of the 21st century, when dramas dominated in this field, many comedies with religious content are now being made (Golubeva 2023).

The issue of defining the genre of religious cinema is up for debate[2]. This article will consider films not only set in monasteries, but also those where the motif of faith plays a significant role. The analysed titles in this paper are also part of the thesis of Marek Sokołowski, who stated that "The religious nature of the film today is determined not by the topic and religious content, but by the issue and the way it is resolved" (Sokołowski 2021, p. 8).

Today, the Orthodox Church, similarly to the Catholic Church, recognises the pastoral potential of cinematography. At the turn of the 20th and 21st centuries, The Holy Archire Council developed and adopted the Basis of the Social Concept of the Russian Orthodox Church. According to Rev. Artur Alexeyuk, the adoption of the document "not only opened a new chapter in the relations between the Church and the state in new socio-political and economic conditions after the collapse of the USSR, but defined the basic directions of the Church's activity in shaping the religious and cultural face of Russia" (Alexeyuk 2020, p. 96). The 14th chapter of the Basis was devoted, among others, to culture. It emphasises the religious roots of culture and its evangelistic role.

> 'The secular culture can be a bearer of the good news. It is especially important in those cases when Christian influence in society weakens or when the secular authorities inter in an open struggle with the Church. (...) Cultural traditions help to preserve and enrich the spiritual heritage in a rapidly changing world' (Osnovy social'noj koncepcii Russkoj Pravoslavnoj Cerkvi 2008)

Accordingly, the Orthodox Church uses cinema in pastoral initiatives, in an attempt to spread the faith and appeal to younger generations especially. On the one hand, this is manifested, for example, in showing support for creators both at the financial and promotional level. Such activities include the Orthodox Encyclopaedia project or Christian film festivals. On the other hand, ROC controls the presented content, thus, wishing to curate the artists in a certain way. The same topics are also used by directors who do not create religious cinema, or rather do not profess a given religion[3]. It happens that such motifs are presented in a way that is not fully consistent with Christian doctrine. In addition, the plots in certain films do not always show a clearly positive assessment of the ROC's condition, which is always met with a harsh reaction from representatives of the Church. Currently, this may also entail legal consequences for creators. In terms of the amendments introduced to the Criminal Code in 2013, there is a paragraph concerning the offence against the feelings of believers (Criminal Code of the Russian Federation, Art. 148), which was previously classified as an offence under the Code of Administrative Offences and was associated with minor consequences (currently, the penalty is up to 3 years imprisonment). In addition, since 2023, out of 23 members, there have been 2 ROC representatives on the board of cinema experts of the Ministry of Culture of the Russian Federation. The task of expert councils is to select films submitted to the Ministry and to decide which projects should receive state funding (Istomina 2023). Ministerial support is not the only possible source of funding. After all, streaming platforms actively create their own independent projects. However, state support can significantly affect the distribution of the film.

## 2. Results

The analysed material includes Russian-language films and series made in the 21st century. Also, through the lens of the achievements of the last 25 years, two films shot after

2022 will be discussed. These titles are diverse in terms of genre—they are feature films, series, dramas, and comedies. They are connected by the discussed issues, focusing on religious content that is conveyed in the visual and linguistic layers and emerges at the level of interpretation. At the same time, the plot is not always consistent with the teachings of the Church. Another criterion is popularity. Based on the data presented on kinopoisk.ru, the largest Russian film database, films were chosen with at least 50,000 ratings or shown at two major film festivals at a minimum. This made it possible to limit the analysed data to films that have received a wide audience response and, thus, left their mark on popular culture.

*2.1. Themes*

In recent years, it seems that the figure of the clergyman has returned to the screen particularly often in both feature films and series. While analysing these films, we can distinguish the most popular themes:

- Yurodivy, or the fool for Christ[4]
- A priest with a turbulent past that he cannot completely break away from
- A converted sinner
- Confrontation of old and new views on spirituality
- The theme of the believer's temptation
- An Anti-Hero, I.E., Negative figures of clergy (especially their greed, pride, jealousy, attachment to wealth)
- A clergyman—wise guide

However, despite the relatively high interest in religious subjects, the portrayal of monastic life is a surprisingly rare occurrence on screen. Among the feature films whose subjects focus on monastic life, it is worth mentioning "The Island" by Pavel Lungin ("Остров", 2006), "The Monk and the Demon" by Nikolai Dostal ("Монах и бес", 2016), and Vladimir Kott's film "Neposlushnik" [Disobedient] ("Непослушник", 2022), analysed here.[5] Lungin's film can be considered a landmark moment for the religious cinema genre in recent Russian cinema. This applies to the portrayal of the monastic community both in the plot plan and the visual layer. "The Island" incorporates most of the above-mentioned motifs. The plot is meant to encourage reflection on sin, self-discovery through the prism of faith, and the acceptance of forgiveness.

"The Monk and the Demon" is also a peculiar novelty for Russian cinema, as it uses the comedy genre to convey religious content, and at the same time, it is inspired by the 12th century life of the Archbishop of Novgorod Ilya. The story of Ivan enslaved by the devil is shown in a perverse way—the actions of the titular bishop, contrary to his intentions, lead to the exposure of sin and conversion. Moreover, the creation of the character of the monk is a direct reference to the attitude of the yurodivy. Ivan, with his eccentric behaviour, notoriously embarrasses, provokes, and throws off the rhythm of everyday life.[6]

In addition, there is almost no mention of nuns or popadias[7] in films[8]. In the last 20 years, only two films that focus on female communities have been shot—"Spring is coming" ("Скоро весна", dir. Vera Storozheva, 2009) and the series "The Monastery" ("Монастырь", dir. A. Molochnikov, 2022). Meanwhile, popadias are most often overshadowed by their husbands, and depicted as women who are benevolent and supporting men in all their activities (e.g., "The Priest" by Vladimir Chotinenko). Against this background, the series "Kidney", directed by Mariya Shulgina, partly stands out ("Почка", 2021) along with the image of popadia Maria presented therein. Although she is a supporting character, it is worth noting that the creation of her character is complex. She appears for the first time during a family gathering. We see her modern and elegantly dressed, although with the inseparable attribute of the wife of an Orthodox clergyman through her headscarf. Maria is

a self-confident, independent woman who decides to live her life. She rebels against her husband, who restricts her freedom and at the same time she is not afraid to oppose or break the conventions. However, it turns out that independence is not Maria's priority in life. When the main character of the series, Nastya, blackmails her by bringing old financial embezzlement to light, Maria humbly returns to her old life. She becomes a model film mother—she changes chic outfits into modest, unfashionable clothes, gives up on makeup, and replaces the pleasures of earthly life with service in the temple and participation in the church choir.

In the latest cinema, the monastery performs various functions. On the one hand, it becomes a safe haven, often as an escape from the turbulent past or the problems faced by the protagonist. In this sense, the monastery is a sacred area, where otherness and detachment from the temporal world is visible. The monastic community is a place of prayer, but also of hard work of its members. Oftentimes, after their initial rebellion and internal disagreement, it is here that the sinner undergoes their fundamental transformation: returning to the path of faith and finding meaning in life. On the other hand, the monastery is sometimes presented as a corrupt place, which is dominated by violence and repressive rule over the individual, most often the superior (hegumen or hegumenia[9]). The analysed films fit into the above categories (Table 1).

**Table 1.** Presence of themes in films presenting monastic life.

| Themes | Films and Series Which Develop a Given Theme |
|---|---|
| Jurodiwyj [holy fool] | "The Island" P. Lungin<br>"The Monk and the Demon" N. Dostal |
| A monk with the turbulent past that affects the present | "The Island" P. Lungin<br>"The Monastery" A. Molochnikov<br>"Spring is coming" V. Storozheva |
| A converted sinner | "The Island" P. Lungin<br>"The Monastery" A. Molochnikov |
| Confrontation of the secular and spiritual world | "The Monastery" A. Molochnikov<br>"Neposlushnik" V. Kott |
| Temptation theme | "The Monk and the Demon" N. Dostal<br>"The Island" by P. Lungin |
| Negative figures of clergy | "The Monastery" A. Molochnikov<br>"The Monk and the Demon" N. Dostal |
| Wise, deeply religious clergyman | "The Monastery" A. Molochnikov<br>"Neposlushnik" V. Kott<br>"The Island" P. Lungin |
| Monastery as a place of refuge | "The Monastery" A. Molochnikov<br>"Neposlushnik" V. Kott<br>"The Island" P. Lungin |

*2.2. Examples*

2.2.1. "The Monastery"

"The Monastery" is one of the few examples of Russian religious series. As Joanna Sosnowska states, "The increasing popularity of the series convention shows that a religious series (with a dominant religious theme) has incredibly many thematic and genre variations and begins to quantitatively catch up with feature cinema" (Sosnowska 2018, p. 266). This trend can also be found in Russia, although the series produced there primarily concern clergy, not monastic communities. "The Monastery" was directed by Aleksandr

Molochnikov at the Kinopoisk film studio. The series consists of six episodes, although work is currently underway on the second season[10]. The show aroused discussions even at the production stage, and the controversy it caused resulted in a record number of viewers for the streaming platform, which exceeded 2.2 million in a month. It should be emphasised that the series, after only two episodes, was met with a negative reaction from the Ministry of Culture of the Russian Federation, which, after consulting with representatives of the Russian Orthodox Church, did not consent to distribution outside Kinopoisk[11]. The decision referred to the law regarding the offence to the feelings of believers. The justification stated that the series shows an image of Russian Orthodoxy that is inconsistent with reality. Above all, it shows a skewed image of monastic life, which may lead to the public forming a false opinion about it (Plamenev and Stogova 2022).

"The Monastery" also attracted attention because of its cast, including the well-known and respected actor Filip Jankowski and the quite controversial Instagram personality and show-woman Anastiasia Ivliejewa, who plays her alter ego. The plot tells the story of bon vivant Maria who chooses a life completely contrary to Orthodox values. The character has nothing to do with religion, since she does not even know if she was baptised. Through her passion for adventure and a lack of moral inhibitions, Maria falls into disfavour with her best friend's husband—an influential billionaire who plans to take personal revenge. Fleeing from her persecutors, the protagonist accidentally ends up in a provincial male monastery, where Father Warsanofij takes care of her and takes her to a nearby female community. The setting clearly determines the presence of visual cues related to religion, but religiosity is manifested primarily at the level of the plot and message of the series. The viewer has a chance to see the gradual transformation of a lost woman who slowly discovers God. Masha's spontaneous behaviour also changes the monks and nuns themselves, bringing a new perspective to their community. This dissonance of two worlds, the contradiction of their values, is already visible in the first few minutes of the series. Parallel scenes show the everyday life in the monastic community—liturgy, hard work, poverty, simplicity, and the rite of monastic haircuts, which are contrasted with the pleasures enjoyed by friends on vacation in the Arab Emirates—romance with accidentally met men, fun in the club, and stimulants. It seems that it is no coincidence that Masha has a golden snake woven into her hair, because she herself becomes a temptress, persuading her friend to reach for drugs for the first time and betray her husband (a parallel with monk haircuts and a juxtaposition of the choice between good and evil).

The second key figure is Father Warsanofij, whose character reflects the image of the clergyman in contemporary Russian cinema quite symptomatically. The first episodes show him as a cordial, loving clergyman who enjoys special respect among his fellow brothers and the faithful. The latter often come to him for advice and help in difficult situations. Warsanofij stands out for his unconventional methods and the ability to look at people in depth. This is well illustrated by the scene where an agitated woman talks about the temptations of the devil, who deceives her to eat during Holy Week. The father responds with care and defiance that temptation concerns only the saints, and the woman simply suffers from hunger, which is why she should eat better, because love for her neighbours is much more important than observing rituals. Furthermore, the protagonist seems not only to accurately diagnose people's problems, but also has the power of healing, for example, he helps a stuttering girl start talking freely.

Warsanofij sees the hidden good in people, as well as hidden sin, and forces the faithful to confront themselves. He is often depicted as absorbed in prayer, from which he draws strength, love, and comfort. It is prayer that allows him to break away from the temptations of worldliness, to cope with the difficulties or baggage of the past. The following episodes show us his previous life, when as the father of two daughters he learns about his wife's

immoral job. Although from the beginning, he is shown as a believer, writing icons and telling children about God, his wife's allusions allow us to assume that he was previously guided by completely different views. The turbulent past can be confirmed by his ability to find a common language with former prisoners helping at the monastery or his advanced knowledge of combat techniques. However, Warsanofij's old life does not haunt him—it is a closed stage for him, which, while returning in memories, does not disturb the peace of his soul. The protagonist looks at the past through the eyes of a deeply religious man.

The women's monastery is opposite to the Warsanofij's community. The main character's daughter was chosen as the hegumenia there. Her behaviour is in direct opposition to her father's values. Instead of love for one's neighbour, she is guided by the rule of power and ruthlessness, instead of wisdom, fanaticism. Therefore, the female monastery becomes a place full of violence, where all power is concentrated in the hands of Jelizaveta and the nuns devoted to her, who are ready to report any misconduct. Hegumenia, hiding behind the concern for the spiritual well-being of her subordinates, forces them to be absolutely obedient in all spheres of life. As it turns out later, these decisions will have dramatic consequences, as they will lead to the death of a young girl.

On the other hand, the figure of the hegumenia does not destroy the sacred character of the place. For many nuns, the monastery remains a place where they feel special closeness to God and fulfil their vocation. One of them, Pantelejmona, assures Masha how much joy it gives her to be in the community, despite the enormous suffering that hegumenia contributes. The sacredness of the place is confirmed by numerous services the women participate in, e.g., the Paschal liturgy.

### 2.2.2. Neposlushniki[12]

Directed by Vladimir Kott (Владимир Константинович Котт), the film did not cause as much controversy as the previously discussed series. As early as the script-writing phase, the creators consulted with clergy from the Shuysk Eparchy of the ROC in order to avoid factual errors in the plot. The artists point out that the recommendations given by the representatives of the Church were only advice and not censorship. Nevertheless, Kott avoids open criticism of the clergy. If the criticism appears in the plot, it takes a rather veiled form.

The director has emphasised that the main purpose of his film was to bring monastic life closer to non-believers. The genre of the film is also not accidental—by using comedy, Kott wanted to show the church as an open place where joy and peace of the soul can be found. The film was supposed to be an invitation for non-religious people (Simankova and Arestova 2024).

The main character, Dmitry, is a man completely detached from spiritual life, and it is only with the course of events that he begins to discover the sacred world. Initially, the viewer gets to know him as a popular youtuber—Dimonstra—who became famous by filming jokes. At the same time, the man generally does not take into account the feelings of his victims. The main determinant is the popularity and approval of the audience, which is why his ideas are cynical and devoid of any morality. In one such video, Dimonstra pretends to be possessed in a church. Consequently, the police initiate proceedings against him, accusing him of desecrating the holy place. In order to escape justice, he flees to his hometown in the province. There, having taken a cassock from his childhood friend, now a clergyman, he begins life in a strict monastery. The conditions in the monastery are extremely difficult—there is no electricity, and water must be brought from a nearby well. A small number of believers and financial problems affect the life of the community. Also, due to the deteriorating technical condition of the buildings, the closure of the entire complex is imminent. Monks are forced to work hard regardless of the season. The film

takes place in winter, so there are scenes of fishing in an ice hole, digging in frozen ground, and cells without heating. It is worth mentioning that placing the monastery in an ascetic winter landscape is particularly popular with Russian directors and has appeared regularly since the time of Lungin's "The Island". Despite the difficult conditions, the monks endure everything with joy and humility of spirit. Hegumen of the monastery is a kind, cheerful old man who gives advice to the faithful. Fellow brothers treat him like a father, trusting his decisions. However, not everyone accepts his attitude—superiors from Moscow criticise his lack of entrepreneurship, and the head of the local government cuts off water from the monastery to punish the clergyman for their lack of political support.

Even though Dmitry initially planned to treat the monastery as a hideout from justice, he undergoes a transformation, similarly to Maria from "Монастырь". Being a self-centred careerist, under the influence of his hegumen, he gradually becomes a person open to the needs of others, seeking love. The scenes of confession are symbolic (apparently a favourite sacrament in films showing clergy). The first one is an opportunity to make another video and gain popularity. The second confession is already taking place in the main setting, and there Dmitry takes the sacrament more seriously, albeit reluctantly. Nevertheless, he opens up and tells hegumen about his problems with honesty. The last confession is an awkward conversation with God, where the protagonist faithfully asks God to restore hegumen's health. Directors use the sacrament of confession to present the spiritual transformation of the hero and show his sinfulness and difficult past, though sometimes the penitent reveals the sinfulness of the confessor with their confession/attitude (e.g., "The Monk and the Demon", "The Island").

Dmitry not only changes himself, but also the monastery—he encourages his hegumen to engage in online ministry, and this idea turns out to be a hit. People are impressed by the simple, yet accurate answers given by the clergyman, and the recordings become surprisingly popular. As a result, a sufficient amount of money is collected for the renovation of the temple and the payment of outstanding bills. In the process, the monastery gains fame, becoming a place of excursions, and the hegumen himself becomes an internet celebrity who young people/influencers want to take pictures with.

## 3. Conclusions

Both films discussed in this article, the series "The Monastery", and the film "Neposlushnik" [Disobedient], may raise objections from film critics, due to the schematics and the use of plot clichés (sinner and saint, conversion of the sinner, etc.). However, in comparison with other titles, they clearly illustrate the tendencies that dominate the world of Russian cinema, presenting the profiles of monks and monastic communities in a fairly convergent way. Similar motifs reappear—the monastery becomes a place of refuge for the heroes and an escape from enemies, and at the same time from the old life. As the heroes cross the monastery gate, a new, groundbreaking stage begins for them. In all the films cited, the directors emphasise the otherness of this place, separating it from the outside world. The dichotomy of these two worlds is marked, among others, by the attitude to the latest technologies, and above all by the values and life priorities. The main characters are outsiders who have to confront their current values with what they encounter inside the monastery. Despite the initial difficulties, they find their place and redefine their own life and vocation (Maria begins a modest life in the countryside, maintaining constant contact with her spiritual father; Dimonstr changes the purpose of his online activity). Simultaneously, the theology of "The Monastery" and "Neposlushnik" is not deep, which may originate from the authors' desire to gain the approval of the widest possible audience. Despite the aforementioned controversies regarding the offences to the feelings of believers in "The Monastery", the world presented in the analysed films is black

and white. Nonetheless, the symbols of religious worship are not used in an objective manner. They interact with the plot elements and together create the religious message of the film. It is worth noting once again that the Russian Orthodox Church has a significant influence on the content presented by the artists. The dynamics between artistic freedom and religious authority in Russian cinema can be seen especially clearly in the example of the series "The Monastery", the continuation of which is likely to be less controversial than the first season. In the context of research on the achievements of Russian cinematography in recent years, the question may arise as to whether Pavel Lungin's "The Island" and Andrei Zvyagintsev's films were the last examples of religious films, drawing on the legacy of Andrei Tarkovsky, and contemporary artists will primarily strive to convey faith-related content in a funny and even superficial way in order to attract the widest possible audience and gain acceptance from the ROC.

**Funding:** This research received no external funding.

**Institutional Review Board Statement:** Not applicable.

**Informed Consent Statement:** Not applicable.

**Data Availability Statement:** No new data were created or analyzed in this study. Data sharing is not applicable to this article.

**Conflicts of Interest:** The author declares no conflict of interest.

## Notes

[1] Compared to the achievements of cinematography of the 1990s and the beginning of the 21st century.

[2] I write more about setting the boundaries of the religious film genre in the monograph: (Kozieł 2021a).

[3] Among them: Kirill Serebriannikov and his "Scholar" ("Ученик", 2016).

[4] In the Orthodox tradition, Jurodiwyjs [holy fools] play a special role. It was the name given to ascetics who, with their unusual, eccentric behaviour, made others reflect on their own faith. Cf. (Ivanov 2005).

[5] The film was not distributed in Western Europe, so I will use the title transcription.

[6] I write more about the portrayal of 'fools for Christ' (yurodivye) in the film "The Monk and the Demon" in the article: (Kozieł 2021b).

[7] Popadia (also: matushka, presbytera)—priest's wife in the Eastern Orthodox (Przyczyna et al. 2022, p. 271).

[8] This appears intriguing in comparison to Western culture, where nuns appear on screen much more frequently, and over the past 20 years, they have been particularly favored by horror filmmakers. Cf. (Dunin-Wilczyński 2023).

[9] Hegumen—head of a monastery in the Eastern Orthodox; hegumenia—head of a convent of nuns in the Eastern Orthodox (Przyczyna et al. 2022, p. 211).

[10] The broadcast is planned for 2025 and the action is to focus on male characters—father Warsanofij and his grandson Jury, who for the first time leaves the walls of the monastery and goes to Moscow. It is worth noting that another director, Sergei Popov, will be responsible for the second season of the series.

[11] It was a severe limitation, as the producers planned to bring the series to the big screen.

[12] At the time of writing, three parts of this film have already been made. They all focus on the figures of Dmitry and father Anatoly. Only the film from 2022 will be used as research material, because the others are based on its plot and feel derivative.

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
