# Peer review of "The Image of Monks and the Monastic Community in the Latest Russian Cinematography"

_religions, doi:10.3390/rel16030351_

Round 1
Reviewer 1 Report
Comments and Suggestions for Authors
This study examines the representation of monastic communities in contemporary Russian cinema, focusing on two post-2022 productions: the series "The Monastery" and the film "Neposlushnik" (Disobedient). "The Monastery" series provoked controversy and was banned from distribution outside streaming platforms, while "Neposlushnik" was produced in consultation with the church, taking a lighter approach to religious themes. This contrast effectively illustrates the current dynamics between artistic freedom and religious authority in Russian cinema. It should be noted that the reviewer has not viewed either of these films.
While religious themes are prevalent in Russian cinema - as the article demonstrates - monastic life remains relatively underrepresented. Monasteries typically serve as places of refuge and spiritual transformation. The research identifies recurring motifs: the converted sinner, conflict between secular and spiritual worlds, and the figure of the holy fool. The Russian Orthodox Church maintains significant influence over film content.
The author has structured the analysis well, providing thorough filmographic background. Their categorization and systematization are clear. They effectively use current, relevant examples and successfully contextualize these two works within Russian film history. They offer comprehensive reflection on contemporary trends.
However, the text is sometimes overly descriptive and lacks a deeper theoretical framework. The author's analysis of the films is brief and could be more profound. Their international comparisons are limited, and the conclusion is rather succinct.
Therefore, I recommend a rating of 2 (Accept after minor revisions). The article provides valuable insight into an understudied topic and effectively documents contemporary trends. With minor revisions (expanding theoretical background, deeper analysis), it merits publication. The author demonstrates thorough knowledge of the subject and ability to systematize observations. After the suggested modifications, the study could serve as a useful resource for researchers in the field.
Author Response
Comment 1: "With minor revisions (expanding theoretical background, deeper analysis), it merits publication."
Lungin's film can be considered a landmark for the religious cinema genre in recent Russian cinema. This applies to the portrayal of the monastic community both in the plot plan and the visual layer. "The Island" incorporates most of the above-mentioned motifs. The plot is meant to encourage reflection on sin, self-discovery through the prism of faith and the acceptance of forgiveness.
"The Monk and the Demon" is also a peculiar novelty for Russian cinema, as it uses the comedy genre to convey religious content, and at the same time, it is inspired by the 12th century life of Archbishop of Novgorod Ilya. The story of Ivan enslaved by the devil is shown in a perverse way - the actions of the titular bishop, contrary to his intentions, lead to the exposure of sin and conversion. Moreover, the creation of the character of the monk is a direct reference to the attitude of the yurodivy. Ivan, with his eccentric behaviour, notoriously embarrasses, provokes and throws off the rhythm of everyday life.
Against this background, the series "Kidney" directed by Mariya Shulgina partly stands out ("Почка", 2021) along with the image of popadia Maria presented therein. Although she is a supporting character, it is worth noting that the creation of her character is complex. She appears for the first time during a family gathering. We see her modern and elegantly dressed, although with the inseparable attribute of the wife of an Orthodox clergyman – a headscarf. Maria is a self-confident, independent woman who decides to live her life. She rebels against her husband, who restricts her freedom and at the same time she is not afraid to oppose or break the conventions. However, it turns out that independence is not Maria's priority in life. When the main character of the series, Nastya, blackmails her by bringing old financial embezzlement to light, Maria humbly returns to her old life. She becomes a model film mother – she changes chic outfits into modest, unfashionable clothes, gives up on makeup, and replaces the pleasures of earthly life with service in the temple and participation in the church choir.
Conclusion:
Both films discussed in this article, the series "The Monastery" and the film "Neposlushnik" [Disobedient], may raise objections of film critics, due to the schematics and the use of plot clichés (sinner and saint, conversion of the sinner, etc.). However, in comparison with the other titles, they clearly illustrate the tendencies that dominate the world of Russian cinema, presenting the profiles of monks and monastic communities in a fairly convergent way. Similar motifs reappear – the monastery becomes a place of refuge for the heroes and an escape from enemies, and at the same time from the old life. As the heroes cross the monastery gate, a new, groundbreaking stage begins for them. In all the films cited, the directors emphasize the otherness of this place, separating it from the outside world. The dichotomy of these two worlds is marked, among others, by the attitude to the latest technologies, and above all by the values and life priorities. The main characters are outsiders who have to confront their current values with what they encounter inside the monastery. Despite the initial difficulties, they find their place, redefine their own life and vocation (Maria begins a modest life in the countryside, maintaining constant contact with her spiritual father; Dimonstr changes the purpose of his online activity). Simultaneously, the theology of "The Monastery" and "Neposlushnik" is not deep, which may originate from the authors' desire to gain the approval of the widest possible audience. Despite the aforementioned controversies regarding the offences to the feelings of believers in "The Monastery", the world presented in the analysed films is black and white. Nonetheless, the symbols of religious worship are not used in an objective manner. They interact with the plot elements and together create the religious message of the film. It is worth noting once again that the Russian Orthodox Church has a significant influence on the content presented by the artists. The dynamics between artistic freedom and religious authority in Russian cinema can be seen especially clearly on the example of the series "The Monastery", the continuation of which is to be less controversial than the first season. In the context of research on the achievements of Russian cinematography in recent years, the question may arise whether Pavel Lungin's "The Island" and Andrei Zvyagintsev's films were the last examples of religious films, drawing on the legacy of Andrei Tarkovsky, and contemporary artists will primarily strive to convey faith-related content in a funny and even superficial way in order to attract the widest possible audience and gain acceptance from the ROC.
Reviewer 2 Report
Comments and Suggestions for Authors
This article seeks to fill a gap in scholarship by analyzing the depiction of monks and monastic life in contemporary Russian cinema. In this respect, the author seeks to do in a scholarly treatment what one of their sources--Anna Golubeva, "'The Disobedient' in 'The Monastery': How Contemporary Russian Cinema Depicts the Church, Priests and Sinners"--does in a popular survey article. The problem is that Golubeva's more breezy essay references with authority more films than does this scholarly treatment. My main suggestion, thus, for improvement is for the author to broaden their treatment in two small but impactful ways: 1) discuss in more detail the important precursor films of this genre that set the cinematic terms of monastery films. In particular, I have in mind here Pavel Lungin's 2006 movie, "The Island" and Vera Storozheva's "Skoro vesna" (2009), which the article mentions but does not analyze for their role in establishing important thematic and cinematic expectations for the depiction of monastery life in contemporary cinema. Both of these two precursors are reflected in the themes of Table 1 but should be examined in more detail before moving on to section two's "example films," "Monastyr'" and "Neposlushnik". 2) I would add a few more examples to the "example films" part of the article besides "Monastyr'" and "Neposlushnik". The article mentions "Monakh i bes" (2016) and "Pochka" (2021) which could be added to bolster their argument and the rubrics all these films suggest. The reader needs assurance that the author has paid sufficient attention to the more important previous and recent films about monastic life.
A smaller detail: the terms "hegumen" and "popadia" should be glossed for those not familiar with them.
I appreciate the list on p. 3 lines 98-106 of the most popular themes in films depicting the clergy. The table on page 6 linking prevalent themes to films is also helpful.
Overall, the article makes a small but distinct and valuable contribution to our understanding of the depiction of religious themes on Russian screens.
Author Response
Comment 1: discuss in more detail the important precursor films of this genre that set the cinematic terms of monastery films. In particular, I have in mind here Pavel Lungin's 2006 movie, "The Island" (…) I would add a few more examples to the "example films" part of the article besides "Monastyr'" and "Neposlushnik". The article mentions "Monakh i bes" (2016) and "Pochka" (2021) which could be added to bolster their argument and the rubrics all these films suggest. The reader needs assurance that the author has paid sufficient attention to the more important previous and recent films about monastic life.
Lungin's film can be considered a landmark for the religious cinema genre in recent Russian cinema. This applies to the portrayal of the monastic community both in the plot plan and the visual layer. "The Island" incorporates most of the above-mentioned motifs. The plot is meant to encourage reflection on sin, self-discovery through the prism of faith and the acceptance of forgiveness.
"The Monk and the Demon" is also a peculiar novelty for Russian cinema, as it uses the comedy genre to convey religious content, and at the same time, it is inspired by the 12th century life of Archbishop of Novgorod Ilya. The story of Ivan enslaved by the devil is shown in a perverse way - the actions of the titular bishop, contrary to his intentions, lead to the exposure of sin and conversion. Moreover, the creation of the character of the monk is a direct reference to the attitude of the yurodivy. Ivan, with his eccentric behaviour, notoriously embarrasses, provokes and throws off the rhythm of everyday life.
Against this background, the series "Kidney" directed by Mariya Shulgina partly stands out ("Почка", 2021) along with the image of popadia Maria presented therein. Although she is a supporting character, it is worth noting that the creation of her character is complex. She appears for the first time during a family gathering. We see her modern and elegantly dressed, although with the inseparable attribute of the wife of an Orthodox clergyman – a headscarf. Maria is a self-confident, independent woman who decides to live her life. She rebels against her husband, who restricts her freedom and at the same time she is not afraid to oppose or break the conventions. However, it turns out that independence is not Maria's priority in life. When the main character of the series, Nastya, blackmails her by bringing old financial embezzlement to light, Maria humbly returns to her old life. She becomes a model film mother – she changes chic outfits into modest, unfashionable clothes, gives up on makeup, and replaces the pleasures of earthly life with service in the temple and participation in the church choir.
Comment 2: A smaller detail: the terms "hegumen" and "popadia" should be glossed for those not familiar with them.
Hegumen – head of a monastery in the Eastern Orthodox; hegumenia – head of a convent of nuns in the Eastern Orthodox (Słownik polskiej terminologii prawosławnej, 2022, p. 211).
Popadia (also: matushka, presbytera) – priest’s wife in the Eastern Orthodox (Słownik polskiej terminologii prawosławnej, 2022 p. 271).